# To Fight or to Grow: The Balancing Role of Ethylene in Plant Abiotic Stress Responses

**DOI:** 10.3390/plants11010033

**Published:** 2021-12-23

**Authors:** Hao Chen, David A. Bullock, Jose M. Alonso, Anna N. Stepanova

**Affiliations:** Program in Genetics, Department of Plant and Microbial Biology, North Carolina State University, Raleigh, NC 27695, USA; hchen22@ncsu.edu (H.C.); dabullo2@ncsu.edu (D.A.B.J.); jmalonso@ncsu.edu (J.M.A.)

**Keywords:** ethylene, abiotic stress, hormone crosstalk, growth and defense tradeoff

## Abstract

Plants often live in adverse environmental conditions and are exposed to various stresses, such as heat, cold, heavy metals, salt, radiation, poor lighting, nutrient deficiency, drought, or flooding. To adapt to unfavorable environments, plants have evolved specialized molecular mechanisms that serve to balance the trade-off between abiotic stress responses and growth. These mechanisms enable plants to continue to develop and reproduce even under adverse conditions. Ethylene, as a key growth regulator, is leveraged by plants to mitigate the negative effects of some of these stresses on plant development and growth. By cooperating with other hormones, such as jasmonic acid (JA), abscisic acid (ABA), brassinosteroids (BR), auxin, gibberellic acid (GA), salicylic acid (SA), and cytokinin (CK), ethylene triggers defense and survival mechanisms thereby coordinating plant growth and development in response to abiotic stresses. This review describes the crosstalk between ethylene and other plant hormones in tipping the balance between plant growth and abiotic stress responses.

## 1. Introduction

Abiotic stresses such as salinity, heat, cold, drought, and flooding have detrimental effects on plant yield and can lead to complete crop failure. In nature, various plants have evolved an array of fascinating mechanisms to rapidly detect changing conditions and respond to the challenge by altering their growth and development, thus enabling adaptation to the unstable environment. The severe phenotypic defects in plant growth and fecundity commonly observed in unfavorable environmental conditions are the consequence of altered developmental programs that plants adopt in an attempt to survive and reproduce under these suboptimal conditions [1]. Stress adaptation usually involves trade-offs between optimal growth and maximal stress tolerance, with stress-tolerant plant varieties typically displaying lower growth rates and productivity under favorable conditions [2,3]. Thus, the focus of modern breeding efforts and biotechnological innovations on minimizing the growth/defense trade-offs and leveraging those natural stress response mechanisms without eliciting a yield penalty will be key to securing a stable future food supply under the ever-changing climate.

One promising factor that may prove crucial to developing effective solutions for uncoupling plant growth from defense is ethylene. Ethylene is an endogenous, gaseous plant hormone that is involved not only in a variety of physiological and developmental processes, from regulating organ growth to inducing fruit ripening, but also in multiple stress responses. In fact, endogenous ethylene production in plants is influenced by several biotic and abiotic factors that affect many of the same physiological and developmental processes that ethylene is known to regulate [4], raising the possibility of rationally manipulating ethylene biosynthesis, signaling, or response to trigger plant adaptation to stress without eliciting growth arrest.

The biological function of ethylene was first characterized in dark-grown pea seedlings that, in the presence of this hormone, display a characteristic set of morphological changes known as the triple response [4]. In the model system *Arabidopsis thaliana* that proved instrumental in the identification of the molecular machinery required for ethylene biosynthesis, perception, and signaling, the ethylene-triggered triple response consists of hypocotyl swelling, exaggeration of the apical hook curvature, and the inhibition of hypocotyl and root elongation in dark-grown seedlings [5]. This robust triple response phenotype has served as the basis for ethylene mutant identification that ultimately led to the cloning and characterization of critical ethylene-related genes [5,6,7].

Ethylene biosynthesis in plants involves three enzyme-catalyzed steps (Figure 1A). Methionine is first converted to S-adenosyl-methionine (SAM) by SAM synthetases, then to 1-aminocyclopropane-1-carboxylic acid (ACC) by ACC synthases (ACS) and, finally, to ethylene by ACC oxidases (ACO) [8,9,10,11]. Ethylene biosynthesis is not specific to any one plant tissue or organ, and it is believed that most, if not all, plant cells can produce ethylene [6]. Under favorable growth conditions, overall ethylene production is kept at a low level, but during specific developmental processes and in response to certain abiotic stresses, ethylene biosynthesis can be rapidly induced [12,13].

Ethylene can freely diffuse through plant membranes reaching the endoplasmic reticulum (ER), where perception takes place and triggers a signaling cascade that ends in the transcriptional regulation of ethylene-responsive genes in the nucleus [14]; Figure 1B,C. The ethylene receptors ETHYLENE RESPONSE SENSOR1 (ERS1), ERS2, ETHYLENE RESPONSE1 (ETR1), ETR2 and ETHYLENE INSENSITIVE4 (EIN4) in Arabidopsis are negative regulators of the pathway that, in the absence of ethylene, activate the CONSTITUTIVE TRIPLE RESPONSE1 (CTR1) protein [15,16]. CTR1 is a kinase that phosphorylates and inhibits the positive regulator of ethylene signaling, EIN2, an ER-localized membrane protein [16]. In the presence of ethylene, upon ethylene binding, the receptors and CTR1 become inactivated, EIN2 gets dephosphorylated and cleaved, and its released C-terminus (CEND) enters the nucleus as well as moves to P-bodies [17,18]. In the nucleus, EIN2-CEND directly or indirectly promotes the activity of EIN3 and EIN3-LIKE1 (EIL1), the transcriptional master-regulators of ethylene signaling. In the cytosol, the EIN2-CEND represses the translation of *EIN3 BINDING F-BOX PROTEIN1 (EBF1*) and *EBF2* transcripts by directly or indirectly binding to their 3′-untranslated regions (3′-UTRs) and moving along with these transcripts to the P-bodies [17,18]. EBF1 and EBF2 are F-box proteins that target EIN3/EIL1 for proteasomal degradation [19,20]. In the presence of ethylene, the translation of *EBF1* and *EBF2* is inhibited, and EIN3 and EIL1 are stabilized and regulate the transcription of ethylene-responsive target genes, such as *ETHYLENE RESPONSE FACTOR*s (*ERF*s) [21,22].

ERFs are AP2-domain-containing transcription factors (TFs) that regulate genes involved in diverse biological processes such as growth, development, hormone, and stress responses through several mechanisms, including transcriptional and post-translational control [23,24,25]. Different subclasses of ERFs have specific DNA binding preferences [26]. For example, a subset of related ERFs recognizes Dehydration-Responsive or C-Repeat Element (DRE/CRT) with A/GCCGAC core sequence found in stress-responsive genes to regulate drought, cold and heat abiotic stress responses. Through direct binding to target gene promoters, ERFs can either activate or repress target gene expression [27,28].

In Arabidopsis, ethylene has been shown to inhibit hypocotyl and root elongation in dark-grown seedlings, but to promote hypocotyl growth in light-grown plants [29]. Thus, by cooperating with different environmental signals, ethylene can differentially regulate growth to mediate plant adaptation to variable environments [30]. Not surprisingly, endogenous ethylene levels and the transcript abundance of ethylene signaling genes are rapidly upregulated in response to multiple abiotic stresses. For example, flooding, heat, shade, heavy metals, salt, osmotic, drought, and cold stress conditions have all been described to trigger ethylene synthesis and increased expression of ethylene pathway genes [31,32,33,34,35,36,37]; Table 1. 

Correspondingly, ethylene presence affects the plant tolerance to these abiotic stresses [56] through multiple levels of regulation (Figure 2). For example, ethylene increases the plant tolerance to salt stress by altering the physiological and developmental activities of plants, e.g., by promoting adventitious root formation, regulating stem and petiole growth, and controlling stomatal aperture [57]. In response to flooding and heavy metal stresses, ethylene cooperates with the reactive oxygen species (ROS) response pathway to modulate plant metabolism and antioxidant system to promote plant survival [58]. Furthermore, ethylene balances photosynthesis and abiotic stress response to fine-tune plant growth and development under unfavorable conditions [59,60]; Figure 2.

To help plants deal with abiotic stress, ethylene engages in a sophisticated crosstalk with multiple other phytohormones to enable plant adaptation [1,31,60,61]. With the explosion in the volume of research studies published in the field of plant biology in the past 30 years, it comes as no surprise that different plant hormones have been found to engage and cooperate in multiple physiological processes, including responses to abiotic stresses [61,62]. Some of this hormone-hormone crosstalk occurs at the level of transcription and involves the coordinated action of regulators for several hormones. For instance, in Arabidopsis exposed to cold stress, JA and GA pathway regulators JASMONATE ZIM-DOMAIN (JAZ) and DELLA proteins physically interact with EIN3 to activate *C-REPEAT/DEHYDRATION-RESPONSIVE ELEMENT BINDING FACTOR* (*CBF*) TF genes to enhance cold tolerance, but *CBF*s can also be induced by ABA [63,64]. This review describes the current state of knowledge on the interactions of ethylene with other plant hormones in response to abiotic stress to balance the trade-off between investing in growth and protecting the plant against stress-induced damage.

## 2. Ethylene and JA

JA is a fatty-acid-derived hormone that was named after *Jasminum grandiflorum*’s jasmine oil [65]. JA was identified for its stress-related functions and the ability to regulate plant growth and development, with its activity being partially dependent on JA interactions with ethylene [66]. JA biosynthesis has recently been reviewed in detail [67,68]. In brief, JAs are predominantly synthesized through two chloroplasts-based pathways: the octadecane route that uses linolenic acid to produce 12-oxo-phytodienoic acid (OPDA) and the hexadecane pathway that converts hexadecatrienoic acid (16:3) to dinor-oxo-phytodienoic acid (dn-OPDA) [69]. These OPDAs are subsequently oxidized into JA in peroxisomes, and the JA is then conjugated with isoleucine (Ile) to form JA-Ile in the cytoplasm. JA-Ile is the bioactive form of JA which regulates plant development and environmental adaptation. JA-Ile binds to the CORONATINE-INSENSITIVE1 (COI1) receptor, inducing the degradation of the JAZ proteins and releasing the MYC2 TF from the JAZ-MYC2 complex. MYC2 then triggers the expression of JA-responsive genes [69].

One of the best-understood points of crosstalk between ethylene and abiotic stress-induced signaling is the transcriptional regulation of stress-related genes by EIN3/EIL1 master-regulators that directly bind to and alter the expression of many TF genes implicated in abiotic stress tolerance [70]. Some examples of such TF genes in Arabidopsis include *ERF1*, *SALT-INDUCED AND EIN3/EIL1-DEPENDENT1*, and *ETHYLENE AND SALT INDUCIBLE1* involved in salt stress response [71], *RELATED TO AP2.2 (RAP2.2)/ERF73* and *RAP2.12* associated with flooding stress tolerance [72], *MEDIATOR COMPLEX SUBUNIT16* (*MED16*) and *MED25* participating in responses to nutrient deficiency [73], *ERF95* and *ERF97* regulating heat stress tolerance [74], and *CBF1, 2* and *3* controlling cold response [51]. Importantly, the transcription of some of the EIN3-regulated downstream TF genes is also controlled by JA signaling. For example, *ERF1* induction in Arabidopsis requires JA under several abiotic stresses, such as drought, salt, and heat stress [75,76]. Through a ubiquitin-mediated de-repression mechanism, JA signaling also regulates EIN3/EIL1 activities. In light-grown Arabidopsis seedlings, JAZ proteins (JAZ1, 3, and 9) can physically interact with EIN3/EIL1 and enhance EIN3/EIL1 binding to HISTONE DEACETYLASE6 (HDA6), a histone tail modification enzyme that blocks transcription [77,78]. The resulting complex that forms in the absence of JA inhibits EIN3/EIL1-mediated transcription [78,79]. Upon JA treatment, JAZ degradation attenuates HDA6-EIN3/EIL1 association, thus activating EIN3/EIL1 [78]. Multiple abiotic stresses enhance endogenous JA production, thereby boosting JA levels in cells and promoting the release of EIN3/EIL1 from JAZ-HDA6 hijacking [78]. Further studies show that EIN3/EIL1 positively regulate JA-dependent root hair development and thus enhance drought tolerance in Arabidopsis [80], whereas JAZ1 and HDA6 are known to, vice versa, inhibit root hair growth and thus attenuate plant drought tolerance in Arabidopsis and rice [81,82,83,84], which is consistent with the cooperative roles of ethylene and JA in regulating plant drought tolerance and root hair development [85].

In addition to the collaborative effects of ethylene and JA on the activity of EIN3/EIL1 and their targets in Arabidopsis and rice seedlings, JA and ethylene can antagonistically regulate many abiotic stress genes in some contexts [86]. In etiolated Arabidopsis seedlings, JA-activated TFs MYC2, MYC3, and MYC4 can physically interact with EIN3/EIL1 and repress their DNA binding ability, thereby reducing the expression of EIN3/EIL1-inducible target genes [86,87]. For example, through this mechanism, the expression of EIN3-regulated flooding stress genes is inhibited by MYC 2/3/4 [88]. Reciprocally, EIN3 can repress the transcription of MYC2-controlled wounding-responsive genes [86]. This mutual inhibition of MYC2 and EIN3 in Arabidopsis modulates the antagonistic effects of JA and ethylene, thereby fine-tuning the expression of many stress-related *ERF*s [75,89,90]. One classical example of the direct target of MYC2 and EIN3 that mediates responses to multiple abiotic stresses is the TF gene *ERF1* [75,89,90] antagonistically regulated by JA and ethylene at the transcriptional level. Arabidopsis ERF1 controls the expression of multiple suites of genes activated by JA, drought, salt, and heat [75,89]. In fact, *ERF1*-overexpressing plants have been shown to have increased tolerance to drought, heat, and salt stress [89].

Besides this layer of transcriptional regulation, JA-triggered inhibition of the EIN3 protein function has also been described to take place at the post-transcriptional level in Arabidopsis apical hook formation [86]. JA treatment decreases EIN3 and EIL1 protein abundance in wild-type Arabidopsis plants, but not in the *ebf1 ebf2* background [86]. Further mechanistic inquiry revealed that JA enhances the transcription of *EBF1*, and this upregulation is diminished in the *myc2* mutant [86]. Taken together, these observations suggest that in Arabidopsis apical hooks, JA triggers EIN3/EIL1 degradation via MYC2-mediated transcriptional induction of *EBF1* [86]. However, it remains unknown if these antagonistic actions of ethylene and JA signaling at the post-transcriptional level also take place in response to abiotic stresses. Furthermore, the stability of EBF1/EBF2 in Arabidopsis is dependent on the RING-type E3 ligase SALT- AND DROUGHT-INDUCED RING FINGER1 (SDIR1), which directly targets EBF1/EBF2 for ubiquitination and proteasome-dependent degradation [91]. Accordingly, SDIR1 promotes drought tolerance and salt sensitivity in Arabidopsis, and the SDIR1-mediated regulation of EBF1/EBF2 is fine-tuned by temperature fluctuations [91,92].

As mentioned above, the cooperative and antagonistic regulation of ethylene and JA crosstalk is largely dependent on the interaction of EIN3, JAZs, HDA, MYCs, and EBFs, whose abundance is tightly controlled by endogenous ethylene and JA. For example, the abundance of EIN3 in Arabidopsis is enhanced by high endogenous levels of ethylene and repressed by high endogenous levels of JA [21,86]. The ethylene and JA production is, in turn, controlled by several abiotic stress factors, with cold stress, for example, reducing the levels of ethylene and enhancing the levels of JA [79]. Accordingly, the abundance and interaction of the aforementioned proteins are altered in response to stress-induced hormone level changes [81,82,83,84,85,86,87]. The relative concentrations of various protein complexes differentially affect plant growth and stress tolerance, a phenomenon illustrated by the JA-ethylene crosstalk mediated by JAZ, HDA, and EIN3 interactions in balancing root hair development and drought tolerance [78,81]. In the future, building the JA- and ethylene-mediated gene regulatory networks from protein-DNA and protein-protein interactions with spatial and temporal resolution in response to specific stress factors would be critical for a comprehensive understanding of the role of ethylene and JA crosstalk in abiotic stress tolerance and plant growth and development.

## 3. Ethylene and ABA

ABA, whose name comes from “abscisin II” [93], is an isoprenoid plant hormone synthesized through the plastidial 2-C-methyl-D-erythritol-4-phosphate pathway. ABA can be produced in nearly all plastid-containing cells and is transported by ABA transporters [94]. In the presence of ABA, soluble PYRABACTIN RESISTANCE1 (PYR1)/PYR1-LIKE/REGULATORY COMPONENT OF ABA RECEPTOR proteins bind the hormone, leading to the inhibition of ABI1-INSENSITIVE1 (ABI1) phosphatase. The ABA-PYR complex causes the accumulation of phosphorylated SNF1-RELATED PROTEIN KINASE2 (SnRK2) family, which in turn phosphorylates various target proteins, including ABA-RESPONSIVE ELEMENT-BINDING FACTORs, to trigger cellular responses [94]. In the absence of ABA, ABI1 inhibits the action of SnRKs, thus switching off the ABA signaling [94].

Abscisic acid is well known as a stress hormone for its role in alleviating abiotic stress in plants [95]. Ethylene and ABA have been described to have antagonistic interactions, influencing each other’s biosynthetic and signaling pathways [96]. Genetic evidence in Arabidopsis suggests that the knockouts of ethylene biosynthesis and signaling genes result in altered plant sensitivity to ABA and, accordingly, affect the ABA-dependent abiotic stress tolerance in several developmental processes [43,46,97,98]. For example, the Arabidopsis *ACS7* loss-of-function mutant with reduced endogenous ethylene levels is hypersensitive to ABA, accumulates higher levels of endogenous ABA, and displays enhanced salt tolerance during seed germination and seedling growth [43,98]. However, the interactions between ethylene and ABA in the regulation of salt tolerance are more complex and developmental processes dependent. For instance, in Arabidopsis leaf development, the loss-of-function mutations of *EIN2* and *EIN3* and gain-of-function mutations of *ETR1* and *EIN4* have been shown to confer salt sensitivity, with these mutants accumulating higher endogenous ABA levels [46,99,100,101,102], whereas the inactivation of *CTR1* and *EBF1/EBF2* leads to enhanced salt tolerance and reduced ABA levels in Arabidopsis [102].

Another abiotic stress-sensitive process affected by the ethylene-ABA crosstalk is stomatal opening and closure (Figure 3). ABA acts as the key regulator of stomatal closure to ameliorate the effect of abiotic stress, such as drought [103]. Exogenous ethylene treatment of different plant species, including Arabidopsis, tomato, carnation, and sunflower, inhibits ABA- and stress-induced stomatal closure [104,105,106,107]. The inhibitory effect of ethylene on ABA-regulated stomata activity stems from flavonol accumulation [108,109,110]. Flavonol build-up under stress conditions is an EIN2-dependent process that represses ABA-induced ROS production and stomatal closure in Arabidopsis [108,109,110]. Controversially, some other studies found that ethylene can cause stomatal closure, resulting from the NADPH oxidase AtRbohF-dependent H_2_O_2_ production through the activation of the Gα protein in Arabidopsis guard cells [109,110], Figure 3. The contradictory results for the effect of ethylene on stomatal opening and closure may be due to differences in experimental conditions, such as stress duration, stress strength, or the measured time point. In an attempt to reconcile these findings and decipher the perplexing control of stomatal closure by the ethylene-ABA crosstalk, a mathematical model that integrates all relevant signaling components in Arabidopsis was constructed for guard cells using a continuous logical modeling framework [111]. This model suggests that an increase in either ethylene or ABA alone results in stomatal closure, whereas the presence of both hormones diminishes stomatal closure.

The complexity in the regulation of stomatal closure via the reciprocal regulation of ABA and ethylene biosynthesis has also been described. Plants overexpressing ethylene-inducible *ERFs* (*AtERF1* in Arabidopsis, *JERF1* in tobacco, *TSRF1* in rice, and *TaERF1* in *Triticum aestivum*) show enhanced expression of ABA biosynthetic genes and greater endogenous ABA content, and thus result in stomatal closure and improved abiotic stress tolerance [75,112,113,114,115]. Reciprocally, high ABA levels in Arabidopsis induce the expression of the TF gene *ABI4*, and ABI4 protein in turn negatively regulates ethylene production by repressing the transcription of ethylene biosynthesis genes *ACS4* and *ACS8* [116], Figure 3. Consequently, high concentrations of ABA limit ethylene production in guard cells and induce stomata closure [117]. The negative feedback loop coordinating the ethylene and ABA levels could partially explain the early and late response of stomata actions in response to abiotic stresses [111], Figure 3.

The antagonistic roles of ethylene and ABA have also been discovered in flooding stress responses. To date, the role of ethylene in flooding stress tolerance has been well-studied in rice and Polygonaceae species whose root systems grow under water submergence [118]. In rice, high concentrations of ABA inhibit shoot elongation [118]. Flooding stress promotes the accumulation of ethylene in rice shoots and that extra ethylene inhibits ABA biosynthesis as well as further decreases ABA concentration by inducing the breakdown of ABA into phaseic acid [119,120,121]. With the degradation of ABA, rapid shoot elongation is promoted [120] and takes plants out of water, thus enabling them to escape from flooding stress.

So far, we have shown that the antagonistic roles of ABA and ethylene are observed in multiple abiotic stress responses. Likewise, ABA and ethylene interact antagonistically in many processes of plant development and growth. For example, ABA negatively regulates seed germination, seedling development, petiole angle, root quiescence center cell division and differentiation, and lateral root emergence, whereas ethylene positively regulates all of these ABA-inhibited processes [122,123,124,125,126,127]. The studies highlight some common downstream regulatory modules, such as TFs, that are antagonistically controlled by ethylene and ABA, and future experimental inquiries would further clarify the nature of this crosstalk and the underlying mechanisms of regulation in balancing plant growth and abiotic stress tolerance. The rich genetic resources available for interrogating ethylene and ABA pathways in model plants Arabidopsis and rice, together with the clear transcriptional roadmaps downstream of ethylene and ABA [128,129], will continue to fuel these important efforts.

## 4. Ethylene and BR

BRs, which were first discovered in an organic extract of rapeseed (*Brassica napus*), are a class of polyhydroxy steroid plant hormones [130]. BRs are synthesized by multiple and complex routes, all of which proceed through triterpenoid pathways from campesterol in the ER [130,131]. BRs can be produced by most plant organs and act locally. BRs bind to the plasma membrane-localized receptor, BR-INSENSITIVE1 (BRI1) in Arabidopsis, enabling BRI1 to interact with its coreceptor BRI1-ASSOCIATED KINASE1 (BAK1). The BRI1-BAK1 complex deactivates another kinase, BRASSINOSTEROID INSENSITIVE2 (BIN2), via a phosphorylation cascade. BIN2 phosphorylates and inhibits multiple TFs, such as BRASSINAZOLE-RESISTANT1 and BRI1-EMS-SUPPRESSOR1. The inactivation of BIN2 by BRs releases and potentiates these TFs to trigger the BR-mediated transcriptional cascade [131].

Similar to ethylene, BR is a hormone that can improve stress tolerance at the expense of altered growth and development. In contrast to the antagonistic role of ethylene and ABA in abiotic stress response, the synergistic interactions between ethylene and BR have been demonstrated in several aspects of plant stress responses in Arabidopsis and tomato [132]. For example, salt stress inhibits seed germination, whereas either exogenous ethylene or BR treatment increases germination rate by ameliorating the inhibitory effect of salt stress in Arabidopsis [133]. Further studies in Arabidopsis found that salt stress reduces ethylene production by repressing *ACO* expression in seed germination, whereas BR reverses the drop in ethylene biosynthesis under salt stress by recovering *ACO* transcript levels [133]. The stimulating effect of BR on ethylene evolution was also observed in tomato seedlings exposed to salt stress [118]. In this system, BR promotes ethylene biosynthesis and signaling by enhancing ACS activity and stabilizing the EILs [134]. The synergistic effect of BR and ethylene then leads to H_2_O_2_ accumulation and improved tomato seedling salt stress tolerance due to an increase in antioxidant enzyme activities [134]. Furthermore, inhibition of ethylene synthesis in cucumber blocks BR-relieved oxidative damage and thus reduces plant tolerance to salt, osmotic, and cold stress, suggesting that ethylene may also be required for BR-mediated abiotic stress tolerance [135]. Likewise, in rice, upon submergence in water, flooding stress induces the expression of an *ERF* gene *SUBMERGENCE1A* (*SUB1A*) which in turn increases BR levels by upregulating the transcription of BR biosynthetic genes. BR and the accumulated ethylene then work together to suppress rice shoot elongation, keeping the plants in a quiescent state and enabling plant survival [136], Figure 4.

Another interesting example that illustrates the synergistic regulatory effects of ethylene and BR is the process of stomatal closure. In Arabidopsis, BR induces stomatal closure by triggering H_2_O_2_ and NO production [110], Figure 3. BR-stimulated H_2_O_2_ and NO accumulation is significantly decreased in the dominant ethylene-insensitive mutant *etr1* but is strengthened in the ethylene overproducing *ACS5* mutant *ethylene overproducer1-1* [110], suggesting that BR-mediated stomatal closure is dependent on ethylene signaling.

Importantly, the synergistic effects of BR and ethylene have been observed not only in abiotic stress tolerance-related processes, but also in normal plant development, such as in root growth or fruit ripening [132]. However, most of the developmental and abiotic stress-related inquiries that explored the BR and ethylene crosstalk in vegetable species lack critical genetic evidence [134,135]. Looking ahead, the systematic analysis of mutants in key hormonal genes in both model and crop species would be necessary to further elucidate the underlying mechanisms of BR and ethylene crosstalk to shed light on the mechanisms that balance plant growth and stress tolerance at the phenotypic and molecular level.

## 5. Ethylene and Auxin

Auxin, whose name originates from the Greek “auxein” meaning “to grow/increase”, was first described to play roles in plant development and growth by a Dutch biologist Frits Warmolt Went [137]. The best-studied type of auxin, indole-3-acetic acid (IAA), is synthesized from the amino acid tryptophan through a two-step pathway. Tryptophan is converted to indole-3-pyruvic acid (IPyA) by the TRYPTOPHAN AMINOTRANSFERASE OF ARABIDOPSIS1 (TAA1)/TAA1-RELATED family, and IPyA is then metabolized to IAA by flavin-containing monooxygenases, YUCCAs (YUCs) [138]. IAA is perceived in the nucleus where it binds to the TRANSPORT INHIBITOR-RESISTANT1(TIR1)/AUXIN SIGNALING F-BOX (AFB) family of auxin receptors. This hormone-binding enables the TIR1/AFB receptors to interact with AUXIN/INDOLE-3-ACETIC ACID (Aux/IAA) proteins, triggering Aux/IAA degradation and the release of AUXIN RESPONSE FACTORs (ARFs) which can now initiate transcriptional cascades of auxin-response genes [139].

The crosstalk between ethylene and auxin has been well characterized in recent years in many developmental processes, especially in root elongation, root hair formation, and adventitious root growth [140,141,142]. In most abiotic stress conditions, including high salinity and water deficit, the developmental plasticity of the plant root is regulated by the auxin-ethylene interaction, and the stress-induced remodeling of root architecture confers stress tolerance [143]. Excess aluminum (Al), one of the major toxic elements in acidic soils, alters the accumulation, polar transport, and the asymmetric distribution of auxin in Arabidopsis roots, leading to reduced root elongation and enhanced Al stress tolerance [144,145,146,147], Figure 5. These shifts in auxin abundance and spatial distribution in response to Al stress arise from changes in the TAA1- and YUC3/5/7/8/9-mediated local auxin biosynthesis [146,147] and PIN-FORMED2 (PIN2)- and AUXIN RESISTANT1 (AUX1)-mediated polar auxin transport in roots [53]. These processes, in turn, depend on proper ethylene biosynthesis and signaling [53,146,147]. Ethylene signaling is thought to act upstream and synergistically with auxin to mediate Al-induced root growth inhibition in Arabidopsis. Furthermore, EIN3 directly activates the expression of *TAA1*, *YUC9*, and *AUX1* to induce auxin accumulation in root transition zones, resulting in the inhibition of root growth [21,53,146,147,148], Figure 5.

Besides Al, other toxic metals characteristic of acidic soils, such as dichromate, cadmium, or excess iron, and alkaline soils have also been described to enhance ethylene production in Arabidopsis roots [148,149,150,151,152]. Plant responses to these toxic metals lead to root growth inhibition and involve enhanced ethylene production and signaling, which in turn induce auxin accumulation and polar auxin transport by upregulating the expression of *AUX1* and/or *PIN2* in Arabidopsis [148,149,150,151,152,153], Figure 5. The resulting root architecture changes triggered by these abiotic stresses in the soil and mediated by the ethylene-auxin crosstalk are thought to increase plant tolerance to these unfavorable conditions [143,152,153].

The auxin- and ethylene-mediated root architecture remodeling also drives plant tolerance in response to nutrient deficiencies [154]. To mitigate the magnesium deficiency stress, plants enhance root hair initiation and growth through the mutual activation of ethylene and auxin signaling [155]. Magnesium deficiency in Arabidopsis induces ethylene production that promotes auxin accumulation by activating *AUX1*, *PIN1*, and *PIN2* transcription in roots. The extra auxin stimulates further ethylene production by inducing the expression of *ACO* and *ACS* enzyme genes [155]. Knockout mutant analyses show that the roles of ethylene in regulating magnesium deficiency-induced root hair development require auxin signaling [155], suggesting that auxin acts primarily downstream of ethylene to regulate root hair morphogenesis. Similarly, ethylene signaling and ethylene-triggered downstream auxin and ROS signaling act together in the inhibition of root hair growth caused by boron deficiency, as evidenced by genetic and chemical treatment analyses [156,157]. Moreover, auxin, ethylene, and ROS also cooperate in regulating root architecture in response to mechanical impedance stress [158].

Another classical example of the ethylene and auxin crosstalk mediating abiotic stress tolerance and leading to major plant architecture and growth pattern changes is the response to flooding stress [159]. Ethylene and auxin interactions play important roles in adventitious root formation, which helps to relieve the flooding stress plants face [159]. In waterlogged *Rumex palustris*, the adventitious root formation is mediated by ethylene accumulation that enhances the sensitivity of root-forming tissues to auxin [160]. In tomato, waterlogging-induced adventitious root formation requires ethylene perception, as revealed by the study of an ethylene receptor mutant, *Never ripe* [161]. Flooding stress promotes ethylene production and, hence, ethylene signaling that, in turn, stimulates auxin transport to submerged stems to induce adventitious root growth [161]. In cucumber, ethylene signal transduction and ethylene-triggered auxin signaling act through the ROS pathway to induce adventitious root growth in response to flooding stress [162]. Together, these studies suggest that auxin signaling works downstream of ethylene to promote flooding stress-induced adventitious root formation in several species.

Besides making adventitious roots, submerged crop species, such as rice and *R. palustris*, often develop aerenchymatous tissue in both roots and shoots to facilitate gas diffusion to the roots to help cope with flooding-induced anoxia [163]. The rice mutant *iaa13* with suppressed auxin signaling shows reduced ethylene-induced aerenchyma formation under O_2_-deprived conditions, along with the decreased expression of ethylene biosynthesis genes [164], supporting the notion that intricate ethylene-auxin interactions are involved in regulating aerenchyma formation in response to flooding/waterlogging stress.

The ethylene and auxin crosstalk is not limited to roots, but also plays a role in hypocotyl responses to heat stress. In Arabidopsis grown at normal temperatures in the light, ethylene-activated EIN3 promotes hypocotyl elongation, in part by stimulating *PIF3* expression [165]. Under elevated temperatures, EIN3 attenuates auxin-stimulated thermomorphogenic hypocotyl elongation through the inhibition of the apoplastic acidification progression by transcriptionally repressing the *APD7* protein phosphatase gene [166], suggesting antagonistic roles of auxin and ethylene in regulating hypocotyl response to heat stress. In contrast, auxin and ethylene show synergistic effects upon heat stress in roots. Elevated temperatures enhance TAA1-dependent auxin biosynthesis through ETR1-mediated ethylene signaling, and the accumulated auxin and subsequent auxin transport are critical for maintaining normal root growth and gravity sensing under heat stress [167].

Taken together, the studies described above highlight the importance of altered root architecture controlled by ethylene-auxin interactions for conferring tolerance to water deficiency, heavy metals, high salinity, nutrient deprivation, and flooding. To further elucidate the mechanisms behind different plant strategies for root architecture remodeling in response to abiotic stress, mathematical models incorporating primary root initiation and lateral root growth governed by the auxin and ethylene crosstalk are required. By quantitatively evaluating the effects of the biosynthetic and signaling components of the individual auxin and ethylene pathways and the genes involved in the crosstalk between these two signals in response to different abiotic stresses, we can gain insights as to how to design and genetically modify plants with enhanced tolerance to these abiotic stress factors while minimizing the tradeoff in plant growth and development.

## 6. Ethylene and GA

GA, whose name originates from the fungus *Gibberella fujikuroi* that induces rapid growth in rice shoots, was first discovered in plants as a growth-regulating hormone [168]. GAs are derived from trans-geranylgeranyl diphosphate, which is first converted to ent-kaurene and then to GA12, a nonbioactive GA. GA12 is then transformed into bioactive GAs, such as GA1, GA3, GA4, and GA7 by the GA20- and GA3-OXIDASE family enzymes [169]. The bioactive GAs in the nucleus bind to the soluble GA receptor GIBBERELLIN-INSENSITIVE DWARF1 (GID1) that interacts with DELLA proteins. When GA-GID1-DELLA complex forms, DELLAs undergo structural changes that enable their recognition by GID2/SLEEPY1 F-box proteins and targeting for proteasomal degradation. With DELLAs degraded, the TFs deactivated by DELLA are released and can initiate the GA-mediated transcriptional cascade [170].

The role of ethylene in plant responses to flooding has been studied in several species. Two distinct ethylene-and-GA crosstalk mechanisms in plants coping with different types of flooding stress have been elucidated to date. In rice, flooding rapidly promotes ethylene accumulation [159]. Upon partial submergence in water, ethylene activates the transcription of class VII *ERFs SNORKEL1* (*SK1*) and *SK2,* and in turn, SK1 and SK2 induce GA signaling-mediated stem elongation in rice [170,171], Figure 4. Some deep-water wild rice varieties elongate their stems through the OsEIL1-mediated induction of a GA biosynthetic gene *SEMIDWARF1* (*SD1*), the gene whose loss-of-function allele in domesticated rice has been at the heart of the Green Revolution [172]. The stem elongation enables rice to escape out from the water and perform photosynthesis to alleviate the flooding stress. With full submergence under water, flooding stress becomes more detrimental for plants, primarily because of limited gas exchange under water, which leads to an energy and carbohydrate deficit. In coping with full submergence under water, rice deprioritizes growth to reach increased hypoxia tolerance. In this way, another rice ERF, SUB1A, with high protein similarity to SKs, is transcriptionally induced by full submergence flooding stress through ethylene signaling, not only dampening ethylene biosynthesis [173,174], but also suppressing GA-mediated stem elongation by directly activating the transcription of DELLA protein genes *SLENDER RICE1* (*SLR1*) and *SLENDER RICE-LIKE1* (*SLRL1*) [175], Figure 4. Furthermore, SUB1A-1 directly activates the transcription of *ERF66* and *ERF67*, whose overexpression activates the expression of anaerobic survival genes and whose protein stability is enhanced under hypoxia stress upon full plant submergence in water [176]. In line with these molecular studies, rice varieties possessing the *SUB1A-1* gene are tolerant to prolonged full flooding [173,174,177]. These two opposite plant responses to partial submergence and full submergences, which rely on different ethylene-induced ERFs and their crosstalk with GA (Figure 4), are exciting examples of how plants can differentially adapt to variable environments to maximize plant survival.

Two contrasting regulatory effects of the ethylene-GA crosstalk on plant growth have also been described in response to osmotic stress. Upon exposure to mannitol, Arabidopsis plants synthesize and accumulate ethylene in actively dividing leaf cells, ultimately leading to the induction of ethylene signaling genes *ETR2*, *ERS1*, *CTR1*, *EIN3*, *EIL1*, *EBF1*, and *EBF2* and several *ERF* genes including *ERF6* and *ERF11* [31]. Studies show that overexpression of *ERF6*, a direct transcriptional target of EIN3 in Arabidopsis [128], not only results in the increased expression of *GA2-OXIDASE6* that degrades GA, but also in the enhanced accumulation of DELLAs due to the stabilization of these regulatory proteins, which in turn inhibit the shoot and leaf growth [177]. Conversely, another mannitol-induced ERF, ERF11, promotes internode elongation and leaf growth by directly increasing GA biosynthesis and, therefore, DELLA degradation and indirectly inhibiting ethylene biosynthesis via transcriptional repression of *ACS* genes [178,179]. Interestingly, both *ERF6*-overexpression lines and *erf11* loss-of-function mutants are more tolerant to mannitol [178,179]. Taken together, these studies demonstrate that ERF6 and ERF11 in Arabidopsis seedlings have opposing roles in regulating the tradeoff between growth and osmotic stress tolerance. Follow-up work has shown that ERF6 acts as an upstream regulator of *ERF11* through direct transcriptional activation [178,180], suggesting that the effect of ethylene and GA interaction may quickly shift in response to abiotic stress.

To sum up, the interaction between GA and ethylene in abiotic stress tolerance is primarily elucidated at the transcriptional level, with the two hormones affecting the same set of TFs. As in the case of JA-ethylene synergy in etiolated Arabidopsis seedlings, GA was also discovered to enhance apical hook curvature, at least in part via a release of EIN3 from the DELLA-EIN3 repressive complex [21,181]. Given that the JA signaling components JAZs and MYCs form repressive complexes with EIN3/EIL1 [78], constructing a transcriptional regulatory network incorporating JA, GA, and ethylene TFs under different types of abiotic stresses would be important to understand how plants cope with adverse environmental conditions by fine-tuning the transcriptional cascade through leveraging the hormone crosstalk.

## 7. Ethylene and SA

SA, whose name comes from the Latin word *salix* (willow), is a phenolic compound produced by plants in response to pathogen exposure and was first identified as a plant immunity-related hormone [182]. SA is synthesized from chorismite through two independent pathways, the isochorismate, and the phenylalanine ammonia-lyase (PAL) pathways. SA binds to NON-EXPRESSOR OF PATHOGENESIS-RELATED1 (NPR1) oligomers and causes the complex to dissociate into monomers that migrate to the nucleus. In the nucleus, NPR1 interacts with the TGACG-BINDING FACTOR family of TFs to induce the expression of *PATHOGENESIS-RELATED* genes [183].

The rapidly induced ethylene production triggered by environmental stress is thought to modulate oxidative stress, which in turn affects plant growth [184]. SA has been reported to inhibit the ethylene production induced by several types of abiotic stresses [185,186]. For example, exogenous SA alleviates heat stress by increasing proline-metabolism and restricting ethylene biosynthesis in heat-stressed plants by inhibiting the ACS activity in wheat [187]. Similarly, the SA-mediated ACS activity inhibition enhances plant tolerance to salt and drought stress in mustard, mung bean, and sweet pepper [188,189,190]. Additionally, exogenous SA has been described to block the conversion of ACC to ethylene, possibly by inhibiting the ACO activity as demonstrated in pear suspension cells [191]. On the contrary, ethylene and SA act in concert to regulate cell death in Arabidopsis and tobacco leaves under ozone (O_3_) stress [192,193]. O_3_-induced ethylene promotes SA biosynthesis by up-regulating the expression of the SA biosynthetic genes encoding CHORISMATE MUTASE and PAL [193]. Inhibition of either ethylene biosynthesis or SA biosynthesis in Arabidopsis rescues the O_3_-induced hypersensitive response cell death phenotype, whereas overproduction of ethylene aggravates cell death [192]. On the other hand, in Arabidopsis, chemical or genetic perturbation of SA signaling blocks ethylene production in response to O_3_, suggesting that SA is required for O_3_-induced ethylene biosynthesis [192].

In summary, many characterized SA-ethylene interactions in abiotic stress responses involve the mutual regulation of these hormones’ biosynthetic pathways. The synergistic effects of ethylene and SA in abiotic stress tolerance propose that SA and ethylene may leverage some common machinery to regulate downstream genes, making this an intriguing possibility to investigate in future studies. Furthermore, given the synergistic nature of the JA-ethylene crosstalk at the transcriptional level and the antagonistic crosstalk between the SA and JA signaling pathways [194], elucidation of the genetic interplay between SA, ethylene, and JA would be another interesting opportunity to explore in the future.

## 8. Ethylene and CK

CKs, whose effects were first uncovered by studying plant shoot induction in tissue culture in response to coconut milk, are involved in the regulation of cell growth and differentiation [195]. CKs are adenine-derived hormones that are synthesized via multiple-step reactions involving ISOPENTENYL TRANSFERASE (IPT), CYTOCHROME P450 FAMILY 735A, and LONELY GUY enzymes [196]. In cells, CKs bind to the receptors ARABIDOPSIS HISTIDINE KINASEs (AHKs) and induce their autophosphorylation. AHKs also phosphorylate ARABIDOPSIS HISTIDINE PHOSPHOTRANSMITTER (AHP) proteins that in turn activate type-B ARABIDOPSIS RESPONSE REGULATOR (ARR) TFs to initiate CK-mediated transcription [195].

Just like in the case of stress-triggered ethylene production, abiotic factors can rapidly modulate CK levels in plants by altering the expression of CK biosynthetic genes [197]. In Arabidopsis, a high level of CK has been described to stabilize *ACS5* and *ACS9* [198,199,200], thus leading to ethylene accumulation. Reciprocally, Al stress-induced ethylene is known to enhance local CK biosynthesis by activating the expression of CK biosynthetic genes encoding IPTs, leading to root growth inhibition under Al stress [148,201], Figure 5. Not only do ethylene and CK mutually regulate each other’s biosynthetic pathways, but also ethylene interferes with the CK signaling output at the level of transcriptional regulation [51]. Specifically, Arabidopsis EIN3 directly suppresses the expression of the CK signaling repressor genes, type-A *ARR*s *ARR5, ARR7*, and *ARR15*, by directly binding to their promoters [51]. Overexpression of these three *ARR*s confers enhanced freezing tolerance [51], suggesting that these genes serve as mediators of the crosstalk between ethylene and CK in the cold stress response [51].

Thus, the bulk of current studies focusing on CK and ethylene interactions in abiotic stress are consistent with the model where CK induces ethylene biosynthesis and, vice versa, ethylene induces CK biosynthesis and signaling. However, these two hormones also intersect in normal plant development. For example, in root growth regulation, the ethylene receptor ETR1 upon binding ethylene was shown to directly activate a multistep phosphorelay CK signal transduction cascade (AHK-AHP-ARRs), integrating ethylene and CK in the control of root apical meristem size [202,203]. It will be interesting to test if the ETR1-mediated CK signaling also plays a role in abiotic stress responses.

## 9. Concluding Remarks and Future Perspectives

In this review, we summarized several lines of evidence supporting the notion that the growth-stress tolerance trade-off is in part regulated by the active crosstalk between ethylene and other plant hormones. The pertinent knowledge acquired in the past few decades about ethylene signaling and its interactions with other hormones will guide scientists in their efforts to engineer better-performing crops in which stress responses are uncoupled from the growth penalty, and high yields are maintained under suboptimal environmental conditions. As described above, some hormone interactions enable plants to enhance the growth of certain plant organs or tissues in response to specific stresses, and such selective growth helps the overall plant survival and ultimate productivity by reducing the adverse effects of stress. The rational strategy would then be to equip crops with such specialized tissue-specific responses that permit plant adaptation to the corresponding types of stress. For example, by modulating the auxin-ethylene crosstalk in rice roots, aerenchymatous tissue formation could be induced to improve crop growth and yield under oxygen-deficient conditions [164,204,205,206]. Similarly, crops with enhanced adventitious roots, lateral roots, and root hairs can be obtained by engineering JA, auxin, or ethylene crosstalk, and these root-boosted plants would have better growth and yield in nutrient-poor soils and under water-scarce conditions, as has been demonstrated in multiple studies, including those performed on tomato [161] and maize [207,208,209]. Alternatively, under some abiotic stress conditions, rational manipulation of hormone crosstalk can deprioritize the plant growth but trigger specialized metabolism and antioxidant systems to enhance plant stress tolerance and survival. Yet, in other scenarios, the hormone crosstalk can maximize organ growth to enable the plant to escape from unfavorable conditions. The seemingly contradictory approach is well-illustrated by the aforementioned ethylene-GA crosstalk-mediated flooding stress response in rice, with opposite strategies adopted in full versus partial submergence conditions (Figure 4). In fact, both strategies have been implemented in transgenic plants to enhance flooding stress tolerance [170,174,210].

Given the great potential of hormone engineering in improving plant growth and crop yield under unfavorable environmental conditions, future efforts should also be directed at bettering our understanding of hormone interactions at the basic mechanistic level. At present, many gaps remain in our comprehension of how the hormone crosstalk is modified by the environment and serves to balance plant growth with abiotic stress responses. For example, while we know that the endogenous levels of ethylene and other hormones in plants are rapidly altered in response to abiotic stresses, the mechanisms that detect and then translate the environmental signals into changes in the biosynthesis or catabolism of these growth regulators are only starting to get unraveled. Hypoxia or flooding stress, for example, could, in theory, be directly sensed by the loss of oxidation of the N termini of proteins, as may be the case for the ACO1 enzyme involved ethylene production and a subgroup of ERFs involved in ethylene responses, which ultimately modulate the plant tolerance to these types of stresses [211,212,213,214,215]. Likewise, heat stress may be sensed by the red/far-red light receptor PHYTOCHROME B [216,217], osmotic/drought stress is perceived by the calcium channel REDUCED HYPEROSMOLARITY-INDUCED [Ca^2+^] INCREASE1 (OSCA1) and its paralogs [218], cold stress is detected by the OPEN STOMATA1 pathway [219], and salt stress is discerned via binding of monovalent cations to the glucuronic acid of the glycosyl inositol phosphoryl ceramide sphingolipids [219]. However, besides the aforementioned flooding stress-induced hypoxia, it remains elusive how this initial perception is mechanistically linked to the changes in hormone biosynthesis and/or signaling and how the coordination between the respective hormonal pathways is achieved. Furthermore, plants in the real world are often exposed to various combinations of abiotic stresses, and it remains to be deciphered how ethylene and other hormones are leveraged by plants to integrate the inputs from multiple stress signals and at what level the signaling cascades intersect and feedback onto one another. With the development of sensitive analytical and genetic methods of hormone quantification and monitoring, as well as various high-throughput omics approaches and mathematical models, complex relationships can now be explored at multiple levels, from protein-protein or protein-DNA interactions to intricate transcriptional regulatory networks. These research directions are expected to provide new insights into the mechanisms of adaptive responses co-regulated by multiple hormones at once.

Another interesting and important aspect of the signal crosstalk that needs to be explored at a much greater depth is the spatiotemporal regulation of hormone interactions. The response of plants to any stress is dynamic and often tissue-specific [220]. Therefore, monitoring the temporal evolution of hormone crosstalk in specific tissues or developmental stages under specific conditions (e.g., in the dynamic regulation of stomatal opening and closure in leaves under drought stress, or time-series documenting the process of lateral root initiation and outgrowth in response to nutrient deprivation) is an important endeavor critical for the future of plant trait improvement efforts. Perhaps, a combination of single-cell omics approaches with the development of new, more sensitive versions of hormone reporters and biosensors, calcium, and ROS sensors, etc. and stacking of these detection devices in model plants to monitor the activity of multiple hormones and second messengers at once will be instrumental for reconstructing the sequence of events from the moment the stress is perceived to the phenotypic outcomes. Translating the gained knowledge to non-model crop plants would greatly impact agriculture.

In conclusion, the network of hormone interactions orchestrates both normal plant development and responses to biotic and abiotic stress. How a handful of growth regulators are leveraged by plants to interpret a combination of internal and environmental inputs and tip the balance between resource allocation towards growth versus defense and survival is a fascinating long-term question that will be key to our ability to maintain and further improve agricultural production under deteriorating climates.

## Figures and Tables

**Figure 1 plants-11-00033-f001:**
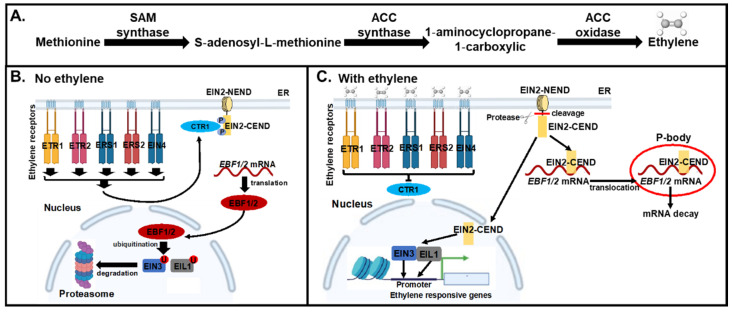
Schematic overview of ethylene biosynthesis and signaling in Arabidopsis. (**A**) Ethylene biosynthetic pathway. Amino acid methionine is converted into hormone ethylene by three families of enzymes, SAM synthetases that produce S-adenosyl-methionine out of methionine, ACC synthases that make 1-aminocyclopropane-1-carboxylic acid, and ACC oxidases that generate ethylene. Ethylene then diffuses out of and into plant cells. (**B**) Ethylene signaling pathway in the absence of ethylene. Ethylene receptors ETR1, ETR2, ERS1, ERS2, and EIN4 localized in the ER membrane activate CTR1 kinase, which in turn phosphorylates the C-terminal end of EIN2 (EIN2-CEND) and turns EIN2 off. The mRNAs for F-box proteins EBF1 and EBF2 are translated and target the master transcriptional regulators of ethylene signaling, EIN3 and EIL1, to proteasomes for protein turnover, thus preventing ethylene responses. (**C**) Ethylene signaling pathway in the presence of ethylene. Ethylene binding to the receptors shuts them off, CTR1 is inactivated, EIN2 is dephosphorylated and is cleaved by an unknown protease, thus releasing EIN2-CEND that functions in the cytoplasm and in the nucleus. In the cytosol, EIN2-CEND recruits *EBF1* and *EBF2* mRNAs to the P-bodies and inhibits their translation. In the nucleus, EIN2-CEND directly or indirectly potentiates the activity of EIN3 and EIL1. Ethylene-triggered stabilization of the EIN3/EIL1 transcription factors leads to the transcriptional regulation of multiple target genes, including induction of several members of the *ERF* gene family that encode transcription factors of the second tier of the ethylene response that propagate the EIN3/EIL1-triggered transcriptional programs. Positive interactions such as activation, production, and stabilization are represented with lines that end in an arrowhead →. Negative interactions such as inactivation and repression are represented by lines that end in a hammerhead ⊣. Images were created with BioRender.

**Figure 2 plants-11-00033-f002:**
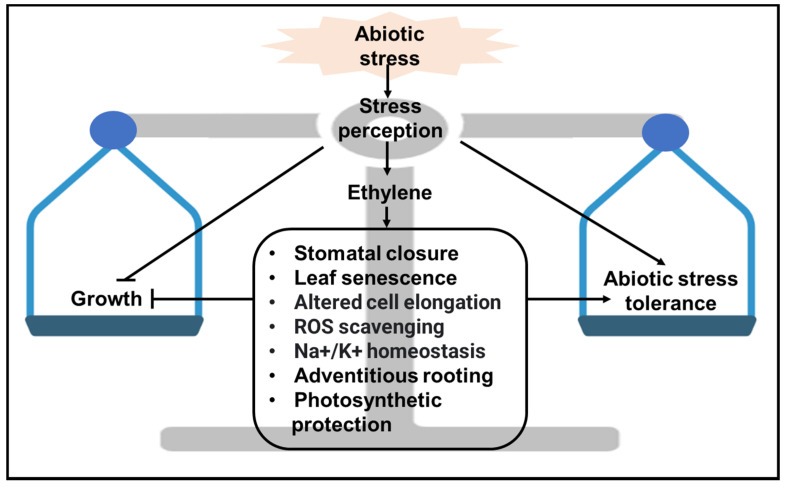
Simplified overview of the role of ethylene in mediating the tradeoff between abiotic stress and growth in plants. Abiotic stress induces ethylene biosynthesis and triggers ethylene accumulation in plants. Ethylene perception and signaling result in multiple physiological responses that not only inhibit plant growth, but also confer stress tolerance that maximizes plant survival in adverse conditions. Positive interactions such as activation, production, and stabilization are represented with lines that end in an arrowhead →. Negative interactions such as inactivation and repression are represented by lines that end in a hammerhead ⊣. The background image of balance was created with BioRender.

**Figure 3 plants-11-00033-f003:**
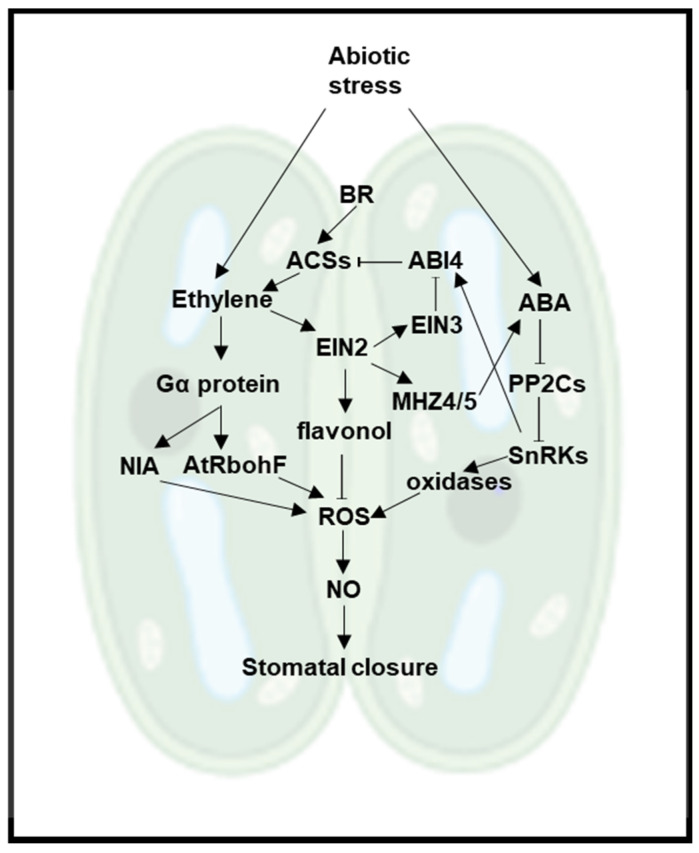
A model of stomatal closure regulation by crosstalk between ABA, BR, and ethylene in Arabidopsis. Endogenous ABA and ethylene accumulate in response to abiotic stress. The biosynthesis of ethylene is upregulated by BR but repressed by ABA due to the transcriptional regulation of *ACS* genes. The ABA biosynthesis is upregulated by ethylene through *MHZ4/5*. ABA induces *ABI4* transcript levels and the ABI4 protein, in turn, represses the transcription of the *ACS* genes. Ethylene also regulates stomatal closure via two ROS-dependent pathways. One pathway is mediated by the ethylene-triggered activation of the Gα protein, which in turn promotes H_2_O_2_ production by activating the NADPH oxidase gene *AtRbohF* and by inducing NO accumulation via the transcriptional upregulation of the nitrate reductase gene *NIA1*. The increased ROS (H_2_O_2_ and NO) results in stomatal closure. The second pathway, vice versa, links ethylene with reduced ROS levels via EIN2-dependent synthesis of flavonols, secondary metabolites that serve as ROS scavengers. Conversely, high levels of ABA signaling lead to elevated ROS production and stomatal closure. Positive interactions such as activation, production, and stabilization are represented with lines that end in an arrowhead →. Negative interactions such as inactivation and repression are represented by lines that end in a hammerhead ⊣. The background image of stomata was created with BioRender.

**Figure 4 plants-11-00033-f004:**
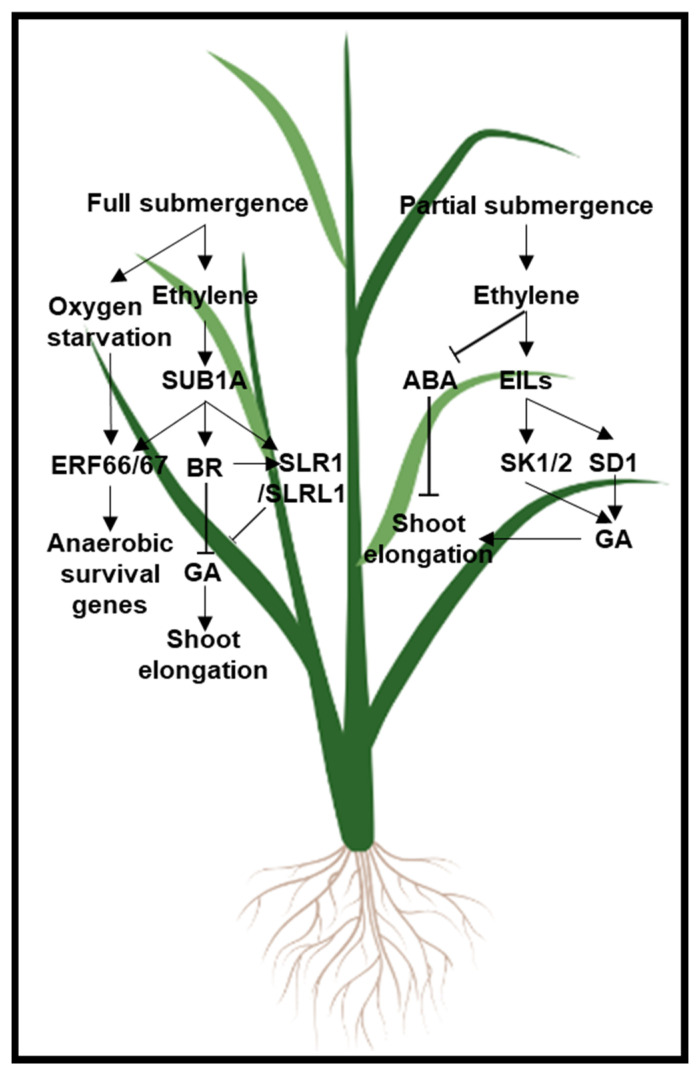
A model of rice adaptation to partial and complete water submergence regulated by GA, ABA, BR, and ethylene. Upon full submergence of rice plants under water (left box), ethylene activates transcription of an ERF TF gene *SUB1A*, and the SUB1A protein then up-regulates anaerobic survival genes via the transcriptional induction and protein stabilization of ERF66/67 upon hypoxia stress. In addition, ethylene suppresses GA-mediated stem elongation by directly activating the transcription of DELLA protein genes, *SLR1* and *SLRL1*, as well as by inhibiting BR signaling. Upon partial submergence under water (right box), ethylene acts via the EIL proteins to induce the transcript levels of *ERF* genes *SK1* and *SK2* bringing about the GA signaling-mediated stem elongation in rice. In deepwater wild rice, EIL-mediated signaling triggered by the hormone ethylene directly enhances GA production and induces shoot elongation by transcriptionally activating a GA biosynthesis gene *SD1* that encodes a GA20-oxidase. Furthermore, ethylene promotes shoot elongation by relieving ABA-imposed shoot growth inhibition. Positive interactions such as activation, production, and stabilization are represented with lines that end in an arrowhead →. Negative interactions such as inactivation and repression are represented by lines that end in a hammerhead ⊣. The background image of a rice plant was created with BioRender.

**Figure 5 plants-11-00033-f005:**
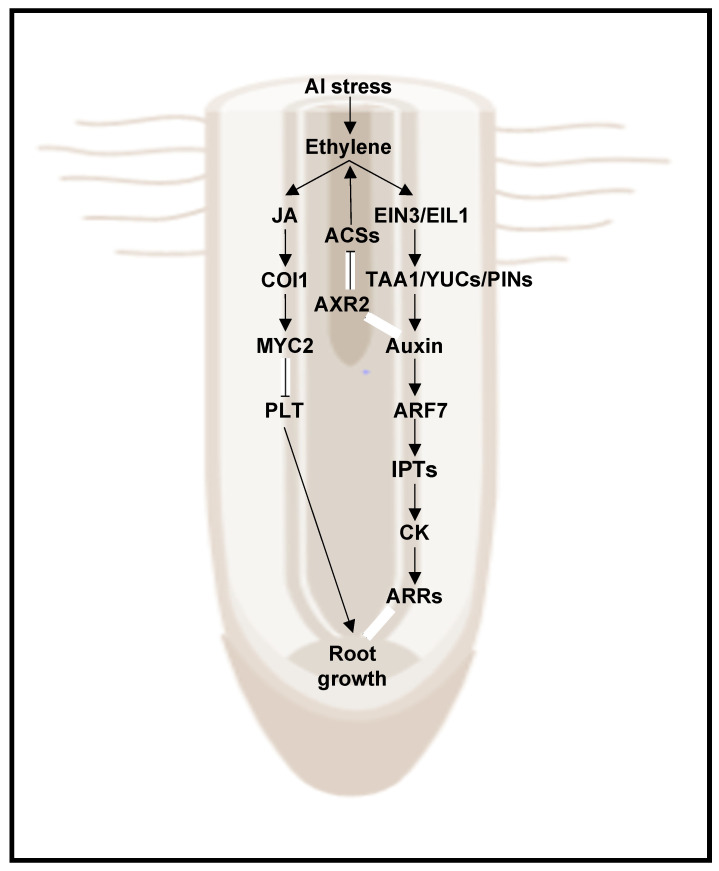
A model of root growth regulation in Arabidopsis triggered by Al stress and mediated by the crosstalk between JA, CK, auxin, and ethylene. Under Al stress, enhanced ethylene production upregulates the local transcript levels of auxin biosynthesis genes *TAA1* and *YUC*s promoted by the transcriptional upregulation of *EIN3* and *EIL1*. The increased protein abundance of TAA1 and YUCs results in local auxin accumulation within the roots, suppressing primary root growth. Downstream of auxin, local accumulation of CK triggered by ARF7-mediated induction of the CK biosynthesis genes *IPT*s activates the CK response, which also contributes to root growth inhibition in response to Al stress. In parallel, COI1-mediated JA signaling downstream of ethylene is involved in Al stress-induced root growth inhibition through the transcriptional repression of *PLT* genes encoding AP2 TFs. Positive interactions such as activation, production, and stabilization are represented with lines that end in an arrowhead →. Negative interactions such as inactivation and repression are represented by lines that end in a hammerhead ⊣. The background image of an Arabidopsis root was created with BioRender.

**Table 1 plants-11-00033-t001:** Effects of different abiotic stress factors on transcript abundance of ethylene biosynthetic and signaling genes in Arabidopsis.

Stress Type	Regulation	Ethylene Biosynthetic Genes	Ethylene Signaling Genes
Drought stress	Up	*ACS8, 12* [37,38]	
	Down	*ACO2, 4, 5, ACS10, 11* [38,39]	
Flooding/hypoxia stress	Up	*ACS2, 6, 7, 9, 11* [40,41]	*EIN3* [42]
	Down		
Osmotic stress	Up	*ACS7, ACO2* [43,44]	*ETR2, ERS1, CTR1, EIN3/EIL1*, *EBF1/2* [44,45]
	Down		*ETR1, EIN2* [46,47]
Salt stress	Up	*ACS2, 6, 7, 11, ACO2, 4* [47,48]	*EIN3* [49]
	Down		*ETR1, EIN2* [45,46]
Heat stress	Up	*ACS6, 7, 8, 10, 11, 12, ACO2, 4* [43,50]	*ERS2, ETR2, EIN3* [50]
	Down	*ACS2, 4, 5, ACO1, 3* [50]	*ERS1, CTR1, EIN2* [50]
Cold stress	Up	*ACS2, 11, ACO2* [41,47]	*EIN3* [51]
	Down		*ETR1, EIN4, EIN2, EBF1/2* [51]
Heavy metal stress	Up	*ACS2, 5, 6, ACO1, 2* [52,53,54]	*EIN2* [55]
	Down		

## Data Availability

Not applicable.

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
