# Peer review of "To Fight or to Grow: The Balancing Role of Ethylene in Plant Abiotic Stress Responses"

_plants, 2021, doi:10.3390/plants11010033_

Round 1
Reviewer 1 Report
The authors provide a very comprehensive review on the vast and widely reported interactions of ethylene with several other plant growth regulators and respective regulatory networks. In my opinion the work is suitable to be published in Plants, and my minor suggestion is that the "Concluding Remarks and Future Perspectives" section should be significantly reduced to focus the main points raised by the authors.
Reviewer 2 Report
Comments on “To fight or to grow: the balancing role of ethylene in plant abiotic stress responses” by Chen etc.
This review is of high quality. First, it has reviewed the most important and up-to-date developments about the role of the crosstalk between ethylene and other hormones in abiotic stress responses. Second, it has included some brief historical background of those hormones. Thus, the review has contributed expert analysis with a consideration of a general readership. Overall, it is a quality review.
Minor points:
- Although Figures 1,2 and 3 have nicely summarised the crosstalk, they mainly demonstrate the pathways of those crosstalk take actions in linear pathways. In reality, the concentrations of those hormones are mutually regulated. The authors have summarised some of those mutual effects in the text. For example, they summarised that “Ethylene and ABA have been described to have antagonistic interactions, influencing each other's biosynthetic and signaling pathways”. However, based on Figure 2, it is not possible to see how ethylene could affect ABA. Similarly, it is well known that the concentrations of auxin and ethylene are mutually regulated, but it is not possible to see that in Figure 3. Since the review focuses on the crosstalk between ethylene and other hormones, the concentrations of hormones are of vital importance. Therefore, it is useful to highlight that the concentrations of those hormones are mutually regulated in Figures 1, 2 and 3 if it is possible.
- Following the title of this review, it focuses on” the balancing role of ethylene in plant abiotic stress responses”. This means the trade off is a focus of this review. Although some texts in the review have discussed trade off, it is not very clear about how ethylene actually plays its role in balancing the trade off. If possible, the authors could include an additional figure (or table, or text) to demonstrate an example about how ethylene balances the trade off under an abiotic stress. For example, an example shows that how a changed concentration/response of ethylene has affected defence and growth in an opposite way. This would highlight the balancing role of ethylene in plant abiotic stress responses.
Comments on “To fight or to grow: the balancing role of ethylene in plant abiotic stress responses” by Chen etc.
This review is of high quality. First, it has reviewed the most important and up-to-date developments about the role of the crosstalk between ethylene and other hormones in abiotic stress responses. Second, it has included some brief historical background of those hormones. Thus, the review has contributed expert analysis with a consideration of a general readership. Overall, it is a quality review.
Minor points:
- Although Figures 1,2 and 3 have nicely summarised the crosstalk, they mainly demonstrate the pathways of those crosstalk take actions in linear pathways. In reality, the concentrations of those hormones are mutually regulated. The authors have summarised some of those mutual effects in the text. For example, they summarised that “Ethylene and ABA have been described to have antagonistic interactions, influencing each other's biosynthetic and signaling pathways”. However, based on Figure 2, it is not possible to see how ethylene could affect ABA. Similarly, it is well known that the concentrations of auxin and ethylene are mutually regulated, but it is not possible to see that in Figure 3. Since the review focuses on the crosstalk between ethylene and other hormones, the concentrations of hormones are of vital importance. Therefore, it is useful to highlight that the concentrations of those hormones are mutually regulated in Figures 1, 2 and 3 if it is possible.
- Following the title of this review, it focuses on” the balancing role of ethylene in plant abiotic stress responses”. This means the trade off is a focus of this review. Although some texts in the review have discussed trade off, it is not very clear about how ethylene actually plays its role in balancing the trade off. If possible, the authors could include an additional figure (or table, or text) to demonstrate an example about how ethylene balances the trade off under an abiotic stress. For example, an example shows that how a changed concentration/response of ethylene has affected defence and growth in an opposite way. This would highlight the balancing role of ethylene in plant abiotic stress responses.
Reviewer 3 Report
Dear Authors,
the topic of this review is current and interesting. Individual chapters are well structured and I welcome a summary of future perspectives. Therefore, I have only minor comments on the manuscript:
- I recommend expanding the chapter on ethylene biosynthesis and supplementing it with an illustrative scheme, as biosynthesis is significantly affected by stress conditions.
- For better readability I would supplement each chapter with a diagrammatic representation (table or figure) summarizing your hypotheses.
